# Ultrasound Diagnostic and Physiotherapy Approach for a Patient with Parsonage–Turner Syndrome—A Case Report

**DOI:** 10.3390/s23010501

**Published:** 2023-01-02

**Authors:** Tomasz Wolny, Katarzyna Glibov, Arkadiusz Granek, Paweł Linek

**Affiliations:** 1Musculoskeletal Elastography and Ultrasonography Laboratory, Institute of Physiotherapy and Health Sciences, The Jerzy Kukuczka Academy of Physical Education, Mikołowska 72A, 40-065 Katowice, Poland; 2Department of Internal Diseases, Rehabilitation and Physical Medicine, Military Medical Academy, Memorial Teaching H of The Medical University of Lodz—Central Veterans Hospital, 90-419 Lodz, Poland; 3Hospital of the Ministry of Interior and Administration, 25-316 Kielce, Poland

**Keywords:** Parsonage–Turner syndrome, neuralgic amyotrophy, ultrasound imaging, physiotherapy, manual therapy, neurodynamic techniques, case report

## Abstract

Parsonage–Turner syndrome (PTS) is a rare neurological disorder that causes major diagnostic problems. This paper presents a case report of a patient with PTS and proposes a new physiotherapy program. Case description: a 23-year-old man presents a sudden severe pain of his right arm. The man is consulted by several doctors and physiotherapists. Three magnetic resonance imagings (MRI), a nerve conduction study (NCS), and needle electromyography (EMG) are performed. After 6 months, based on medical history, physical examination and ultrasound imaging (UI), the physiotherapist suggests PTS, which is confirmed by a neurologist. Intervention: due to the lack of physiotherapy treatment standards in PTS, we apply neurodynamic techniques. Outcomes: neurodynamic techniques are effective in reducing pain and paraesthesia, improving sensation, and reducing nerve swelling (assessed by UI), as well as improving manual dexterity and overall health status. Conclusions: the patient with PTS is challenging for making an accurate diagnosis. This study shows an important role for UI, which shows changes in the musculocutaneous nerve, despite the lack of abnormalities in the MRI, NCS, and EMG, and helps in making an accurate diagnosis. This report also confirms that physiotherapy based on neurodynamic techniques may have beneficial effects in PTS.

## 1. Introduction

Parsonage–Turner syndrome (PTS), also known as neuralgic amyotrophy (NA), is a rare neurological disorder of unknown aetiology [1,2]. The overall prevalence of PTS is estimated at 1.64–3.00 cases per 100,000, but this rate may be underestimated because of major diagnostic problems [1,3]. PTS usually affects men more often than women and is most commonly diagnosed between 30 and 70 years of age [1,3,4,5]. The possible causes of PTS include immunological, environmental, and genetic factors [1,6,7].

PTS is characterised by the acute onset of severe pain that is localised to the shoulder girdle, arm, or elbow, with frequent radiation to the neck region/or the entire upper limb, which usually persists from several days to several weeks [6]. With time, the pain starts to reduce, but muscle weakness and atrophy of the muscles innervated by the various nerves of the shoulder plexus occur [1]. In PTS, sensory disturbances (hypaesthesia, paraesthesia, and allodynia) are also commonly presented [8]. Briefly, typical PTS nature is described as acute painful onset and quick muscle atrophy, which can last for many years and impair the function of the upper limb [9]. In clinical practice, diagnosis of PTS is a diagnosis of exclusion by a process of elimination of other conditions because pain in the area of the shoulder girdle and upper limb may also have other causes that need to be taken into account in differential diagnosis [10,11].

Ultrasound imaging (UI) is a promising tool in the diagnosis of peripheral neuropathies [12] including the most common upper limb neuropathies, such as carpal tunnel syndrome [13] or cubital tunnel syndrome [14]. The most common UI assessment is the measurement of the cross-sectional area (CSA) of the nerve, as the nerve tends to flatten at the site of compression and swell proximally to the compression site [14,15]. Therefore, it is likely that the use of UI may also be useful in the diagnosis of PTS.

In the available literature, there is a lack of studies and recommendations on PTS treatment [1,6,7]. In most studies, symptomatic treatment with analgesics, nonsteroidal anti-inflammatory drugs, and corticosteroids is used to relieve pain [1,7]. Surgical treatment of people with PTS is controversial and not widely used [7]. Some authors have also emphasised the crucial role of physiotherapy in PTS management, intended to alleviate symptoms of the condition [1,2,7]. Unfortunately, we have found only one article with a detailed description of physiotherapy in PTS [16]. Other articles have only presented a general overview of physiotherapy goals in PTS [2,7].

Our prior studies have confirmed the effectiveness of neurodynamic techniques in carpal tunnel syndrome [17,18,19]. The effectiveness of neurodynamic techniques was also confirmed in other peripheral neuropathies [20]. In theory, application of neurodynamic techniques should restore the dynamic balance between the nerve and surrounding tissues, and then improve the physiological function of the nerve. The potential benefits of neurodynamic techniques in peripheral neuropathies are related to reduction in oedema and adhesion, improvement of blood supply and nerve nutrition, improvement of axonal transport and nerve conduction, and reduction in ischemia-related pain [17]. Additionally, our positive clinical experience of using neurodynamic techniques in other peripheral neuropathies (unpublished data) has prompted us to apply such a therapy in PTS. To the best of our knowledge, neurodynamic techniques have never been used in PTS treatment. Thus, this paper presents a chronological case report of a patient suffering from severe pain in the elbow area with subsequent atrophy of the brachialis muscle, who finally received a PTS diagnosis. This paper also shows the usefulness of UI examination in making an accurate PTS diagnosis. In addition, this case report presents a detailed physiotherapeutic intervention based on neurodynamic techniques for the conservative treatment of PTS. We believe that this study may help other clinicians and researchers in the development of PTS treatment standards.

## 2. Case Description

This case study presents a 23-year-old (182 cm height, 70 kg weight, BMI = 21.13) right-handed male (a student of medicine). The man was previously healthy, with no chronic diseases and no history of genetic or inherited disorders in the family. During the interview, the patient reported an active lifestyle and intensive exercise at the gym in his spare time. The diagnostic procedure and symptomatic pharmacological treatment lasted from May to November 2021. During this time period, the patient visited several doctors and physiotherapists. Two ultrasound imaging examinations (UI), three magnetic resonance imaging (MRI) scans, a nerve conduction study (NCS), and electromyography (EMG) were performed. Several different diagnoses were considered. After six months of tests performed and as a result of gradual exclusion of other potential causes, the final diagnosis based on medical history, physical examination, and UI was confirmed. It should be emphasised that the MRI, NCS, and EMG examinations did not show any significant changes that might indicate PTS. The UI examination turned out to be only one sensitive, in that it showed swelling of the musculocutaneous nerve. A detailed description of the case is presented in Table 1. We checked the entire manuscript using the CARE checklist for case studies to ensure that it contained all the guidelines necessary for this type of work (Appendix A).

**Table 1 sensors-23-00501-t001:** A detailed description of the patient’s case.

May	Symptoms: severe, throbbing, and diffuse pain in the biceps muscle of his right arm (pain 8/10 NPRS). Assessment: medical history (family doctor). Diagnosis: exercise-related overload at the gym. Treatment: pharmacological analgesics.
June	Symptom: increasing muscle weakness (pain 6/10 NPRS). Assessment: physical examination (physiotherapist). Diagnosis: overload of the biceps muscle. Treatment: cold compresses. Effects: no positive therapeutic effect.
July	Symptoms: brachialis muscle atrophy (Figure 1 and Figure 2) (pain 4/10 NPRS). Assessment: medical history, physical examination, ultrasound (orthopaedic surgeon). Diagnosis: unspecified soft tissue diseases associated with their use overload and overexertion (M70.9 ICD). Treatment: pharmacological treatment, avoid full weight-bearing. Effects: no positive therapeutic effect.
August	Symptoms: further brachialis muscle atrophy, worsening of pain after exercise (pain 8/10 NPRS). Assessment: no assessment. Diagnosis: no diagnosis. Treatment: patient’s decision: nonsteroidal anti-inflammatory drugs (pain reduction 4/10 NPRS) Effects: pain reduction (4/10 NPRS).
September	Symptoms: further brachialis muscle atrophy, worsening of pain after exercise (pain 8/10 NPRS). Assessment: no assessment. Diagnosis: no diagnosis. Treatment: patient’s decision: nonsteroidal anti-inflammatory drugs (pain reduction 4/10 NPRS) Effects: pain reduction (4/10 NPRS).
October	Symptoms: episodic of more severe pain, paraesthesia, tingling, numbness and burning sensations of right upper limb, (pain 3/10 NPRS). Assessment: medical history, physical examination (another orthopaedic surgeon), shoulder (Figure 3) and elbow MRI order (the same orthopaedic surgeon), cervical spine MRI order, electromyography, nerve conduction study (Table 2 and Table 3) (another orthopaedic surgeon), medical history, physical examination, ultrasound (another physiotherapist). Diagnosis: mononeuropathy of the upper limb, unspecified (G56.9 ICD) (orthopaedic surgeon), PTS suggestion (physiotherapist) Treatment: orthopedy surgeon: physiotherapy (massage, muscle relaxation, electrostimulation, exercises); physiotherapist recommendation: neurodynamic techniques. Effects: no positive effect.
November	Symptoms: no new symptoms, (pain 3/10 NPRS). Assessment: medical history, physical examination and tests performed to date (neurologist). Diagnosis: confirmation of the diagnosis of PTS. Treatment: neurologist: pharmacological treatment. Effects: No positive therapeutic effect (4/10 NPRS).

NPRS—numeric pain rating scale.

## 3. Physiotherapy Management

During the diagnostic process, the patient was subjected to pharmacological treatment. The initial treatment included nonsteroidal anti-inflammatory and analgesic drugs and dietary supplements containing proteolytic enzymes, collagen, and chondroitin sulphate. It was also recommended to apply cooling compresses and avoid putting too much weight on the limb. Once the right diagnosis was provided, drugs affecting the nervous system (acetylcholinesterase inhibitor) and vitamin supplements were prescribed. Unfortunately, such treatment was not effective. Consequently, the patient went to the same physiotherapist who had previously suggested PTS. The physiotherapist, after re-examining the patient, proposed physiotherapy based on neurodynamic techniques of the upper limb nerves. At the time of the physiotherapy intervention, the patient was not taking any other treatment (including any pharmacotherapy). The patient was informed that the proposed therapy was not supported by any scientific studies and there are no specific recommendations in the scientific literature on PTS treatment. The patient gave written consent to apply this form of therapy and then to describe this case in a scientific journal. No bioethics committee approval was applied to the study, as the present case study was retrospective in nature.

### 3.1. Baseline Assessment

Prior to physiotherapy, an interview was conducted. Pain was assessed by NPRS (numeric pain rating scale) where “0” is no pain and “10” is the most severe pain [21]. Static two-point discrimination in the innervation area of the musculocutaneous nerve was assessed using a calliper according to the methodology mentioned elsewhere [22,23]. Smaller spacing between calliper spikes means better two-point discrimination sensation. A horizontal cross-section area of the musculocutaneous nerve was assessed using UI (Figure 4). The thickness of the right brachialis muscle 3 cm from the bottom of the olecranon fossa was measured by UI (Figure 5). The disabilities of the arm, shoulder, and hand (DASH) [24], which assesses the ability of a patient to perform certain upper extremity activities, and the 36-item short form health survey (SF-36) [25], which assesses overall health status, were also collected.

### 3.2. Intervention

Physiotherapy was administered after the medical doctor’s diagnosis and continued for five months with a weekly frequency. No other form of therapy was used at the intervention time. The patient was allowed to exercise at the gym, but activities that aggravated the pain were not allowed (during the intervention the exercises the patient was doing were not controlled). The physiotherapeutic intervention included following neurodynamic techniques:Median nerve neurodynamic technique (Appendix A):
Neurodynamic sequence: arm adduction to 90°, arm external rotation, wrist and fingers extension, forearm supination, and elbow extension.Tension techniques were performed in the proximal and distal directions: 1-direction proximal tension mobilisation (movement—elbow extension—small amplitude of motion at the end of the movement), and 1-direction distal tension mobilisation (movement—wrist extension—small amplitude of motion).
Ulnar nerve neurodynamic technique (Appendix A):
Neurodynamic sequence: arm adduction to 90°, arm internal rotation, wrist and fingers extension, forearm pronation, and elbow extension.Tension techniques were performed in the proximal and distal directions: 1-direction proximal tension mobilisation (movement—shoulder depression—small amplitude of motion at the end of the movement), and 1-direction distal tension mobilisation (movement—wrist adduction—small amplitude of motion).
Radial nerve neurodynamic technique (Appendix A):Neurodynamic sequence: arm extension, arm adduction to 45°, arm internal rotation, wrist and fingers flexion, forearm pronation, and elbow extension.Tension techniques were performed in the proximal and distal directions: 1-direction proximal tension mobilisation (movement—elbow extension—small amplitude of motion at the end of the movement), and 1-direction distal tension mobilisation (movement—wrist flexion—small amplitude of motion).
Musculocutaneus nerve neurodynamic technique (Appendix A):
Neurodynamic sequence: arm extension, arm adduction to 45°, arm internal rotation, wrist ulnar deviation and thumb flexion, forearm intermediate, and elbow extension.Tension techniques were performed in the proximal and distal directions: 1-direction proximal tension mobilisation (movement—elbow extension—small amplitude of motion at the end of the movement), and 1-direction distal tension mobilisation (movement—wrist ulnar deviation—small amplitude of motion).


Protocol consisted of three series of 60 repetitions of tension neurodynamic techniques separated by interseries intervals of 15 s, once a week. The approximate duration of each session was about 30 min. The starting positions and end positions of the neurodynamic techniques are shown in Appendix A).

### 3.3. Outcomes

Prior to the intervention, we collected a medical history of reported pain and paraesthesia occurring along the arm, forearm, palm of the hand, and up to the thumb. Over the course of the therapy, the patient reported a gradual improvement (pain, paraesthesia, tremors, and myoclonus reduction). Before each session, the patient was asked about their symptoms and well-being in the period between visits and during and after therapy. During and after therapy, the patient did not mention any side effects.

Ultimately, 20 therapy sessions were performed over a five-month therapy period. A detailed re-examination was performed when the pain was completely relieved (after 5 months) during normal functioning, and during and after physical activity (gym exercises). There was also complete elimination of paraesthesia, and the feeling of fatigue and discomfort, which previously appeared after intense physical activity. After intervention, there was also an improvement in two-point discrimination sensation (reducing the distance between calliper spikes), USI measurement of a cross-section area of musculocutaneous nerve (reducing the cross-sectional area). There was also a significant improvement in the functional performance of the hand as evaluated by the DASH questionnaire (increasing the number of points), as well as an improvement in the assessment of overall health status (increasing the percentages of individual components of overall health). No increase in thickness of brachialis muscle was achieved. The detailed pre- and post-intervention results are shown in Table 4. Six months after the end of physiotherapy, in a phone conversation, the patient indicated the absence of any pre-existing symptoms and a full return to daily activities.

## 4. Discussion

This case report had some main purposes. The first purpose was to present atypical symptoms of a PTS patient involving musculocutaneous nerve and atrophy of the brachialis muscle. The second purpose was to show the usefulness of the UI examination in the diagnosis of PTS. The third purpose was to present physiotherapy intervention based on neurodynamic techniques in PTS management. This study presented a young man with challenging PTS symptoms, in which the time between the first symptoms and the diagnosis was 6 months. Such a time span from the first symptoms to proper diagnosis may have a significant impact on symptoms, disability, and initiation of treatment. Thus, PTS patients with heterogeneous and atypical symptoms should be widely presented to help clinicians making an accurate and prompt diagnosis. In the available literature, there is also limited information on physiotherapy procedures in PTS. Thus, it was decided to use neurodynamic techniques as a physiotherapy approach because previous studies confirmed that neurodynamic techniques are effective in NCS, pain reduction, improvement of various types of sensation, overall quality of health, and functional status [17,18,19,20] in some peripheral neuropathies. In the case report presented here, neurodynamic techniques were well-tolerated and no side effects were reported. It seems that neurodynamic techniques have also positively affected many of the PTS symptoms. The pain complaints have gone and paraesthesia and myoclonus feelings of fatigue in the limb were eliminated. Two-point discrimination sensation also improved. The cross-sectional area of the musculocutaneus nerve decreased considerably, indicating a reduction in the nerve’s swelling. There were also improvements in hand function and overall health status.

The diagnosis of PTS is based primarily on clinical history, properly conducted symptomatology, physical examination, and muscle testing [7,26]. It can also be confirmed by clinical neurophysiological studies [27]. Some authors clam that during PTS diagnosis an excessive number of tests, which are very often expensive and inconvenient for patients, are used [28,29]. The patient in this report underwent three MRI scans (shoulder, elbow, and cervical spine), two UI examinations, electromyography, and a nerve conduction study, and was referred for a histopathological examination of the affected muscle. Dill-Macky et al. [30] have suggested that MRI of the shoulder girdle region is not very sensitive, whereas Lieba-Samal et al. [31] emphasised the usefulness of high-resolution UI in the diagnosis of PTS. In fact, our case report has shown the swelling of the musculocutaneous nerve without significant changes in MRI. Therefore, we can confirm the usefulness of UI in the assessment of musculocutaneous nerve oedema and in facilitating the accurate diagnosis of PTS. Many authors emphasise the usefulness of UI also in other peripheral neuropathies of the upper limb [15,32]. In addition, UI can also be used to assess muscle atrophy (brachialis muscle) and track the effects of the therapy. Although NCS is often referred to as the “gold standard” in the diagnosis of peripheral neuropathies [33], nerve conduction parameters in PTS may be within normal limits, as it was shown here. A case report and a literature review by Duman et al. [34] have indicated that magnetic resonance neurography (MRN) has good efficacy in the diagnosis of PTS. They recommended MRN in the diagnosis of acute PTS. Unfortunately, the MRN is little known and rarely performed. The lack of specific diagnostic tools and the insufficient knowledge of many clinicians regarding this condition seem to be the main problems in the diagnosis of PTS. From this perspective, case reports presenting PTS patients are warranted.

For patients with diagnosed PTS, there are no specific treatment algorithms. Some authors [1,2,7] have indicated that physiotherapy is useful for PTS treatment, but without any specific recommendations. There are only general physiotherapy goals (maintaining joint range of motion, improving circulation, and increasing muscle mass and strength) mentioned. It is only suggested that heat, massage, electrostimulation, strengthening exercises, and education may be apply in PTS treatment [1,2,7]. A detailed physiotherapy program was only described in one case report of a patient with PTS [16]. The intervention included postural correction, and taping and strengthening exercises, and was applied for 5 months [16]. After the 5-month physiotherapy program, the patient was pain-free and regained full range of motion in the shoulder joint [16]. In contrast to De Burca’s [16] physiotherapy protocol, we have decided to use neurodynamic techniques in our patient treatment. This decision was due to the dominance of other symptoms and the different localisation of PTS in our patient compared to the De Burca’s patient. Our patient had many neurological symptoms in the upper limb (these symptoms were not present in the patient described by De Burca). Thus, it was decided that neurodynamic techniques may provide a more favourable therapeutic effect than other modalities. Neurodynamic techniques have been successfully used in other disorders with associated neurological disturbances [17,18,19,35,36]. We believe that improvement of our patient was related with neurodynamic techniques that may reduce intraneural oedema (therefore, the cross-sectional area of the musculocutaneus nerve decreased after intervention). This improvement may also be explained by improved blood circulation in the musculocutaneus nerve and the slow and rapid transport of axoplasm within the upper limb nerves and the shoulder plexus. The obtained beneficial effect of the applied neurodynamic techniques seems to justify their further evaluation in patients with PTS. It should also be emphasised that the lack of effect related to the increase in muscle mass and strength resulted from the lack of use of exercises increasing muscle mass and strength at this stage of therapy because of the dominant pain symptoms.

In the present study, as well as in the study by De Burca [16], a similar 5-month duration of intervention was used with a frequency of 1 × per week. De Burka did not justify the ongoing 5-month therapeutic procedure. In our study, the therapy was carried out until the patient experienced a noticeable effect. Fortunately, this corresponded with the elimination of symptoms. We also cannot exclude the possibility that the results obtained are merely the consequence of the passage of time (spontaneous healing process). This is the main study limitation. Ibrahim et al. [37] claimed that although physiotherapy improves and alleviates PTS symptoms, it does not improve recovery duration. Tsairis et al. [4] reported that a recovery in 80% of patients with PTS occurs after two years, and 89% after three years. In the case of our patient (6 months of diagnosis and 5 months of therapy), recovery was after 11 months. Thus, it can certainly be stated that the physiotherapy intervention used had no adverse effects, as our patient’s condition has gradually improved since the start of physiotherapy.

## 5. Conclusions

The present case report revealed a variety of atypical PTS symptoms that were challenging for clinicians. The diagnostic procedure, as well as the long time from symptom onset to diagnosis, indicates the need for further education of clinicians to better understand the symptoms of PTS. Additionally, we have confirmed that neurodynamic techniques seem to be useful in PTS patients with neurological disturbances. Thus, neurodynamic techniques of upper limb nerves can be recommended in conservative treatment of PTS patients. It should be emphasised that the muscle mass of the brachialis muscle did not increase, but the exercises aimed at increasing muscle mass were not used. This case study has also shown an important role of peripheral nerves UI. UI showed changes in the musculocutaneous nerve, despite the lack of abnormalities in the MRI, NCS, and EMG examination.

## Figures and Tables

**Figure 1 sensors-23-00501-f001:**
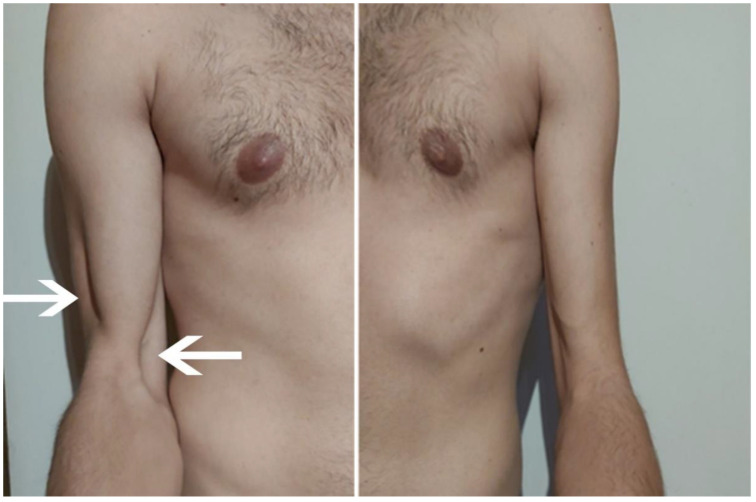
Shoulder on the right and left sides. Significant atrophy of the brachialis muscle is seen on the right side (source: personal).

**Figure 2 sensors-23-00501-f002:**
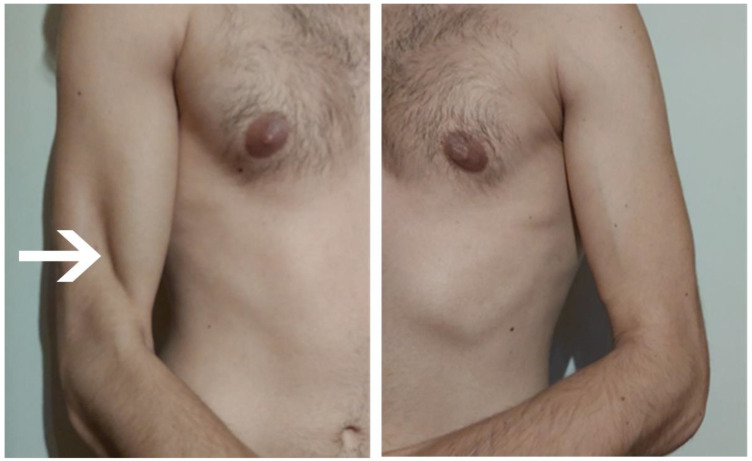
Shoulder on the right and left sides. Significant atrophy of the brachialis muscle is seen on the right side (source: personal).

**Figure 3 sensors-23-00501-f003:**
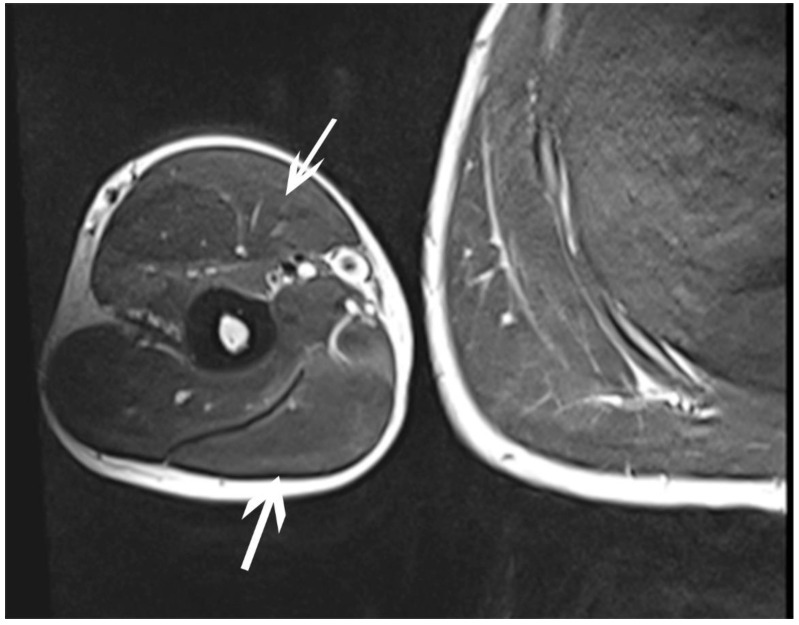
Oedematous changes within the right biceps and triceps brachii muscles—transverse section (source: personal).

**Figure 4 sensors-23-00501-f004:**
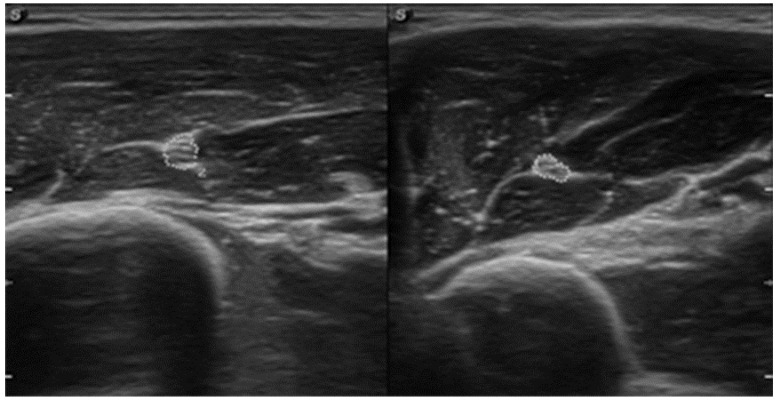
Ultrasound image of a transverse section of the brachialis muscle and cross-section area of the musculocutaneous nerve on the affected (**right**) and non-affected (**left**) side (source: personal).

**Figure 5 sensors-23-00501-f005:**
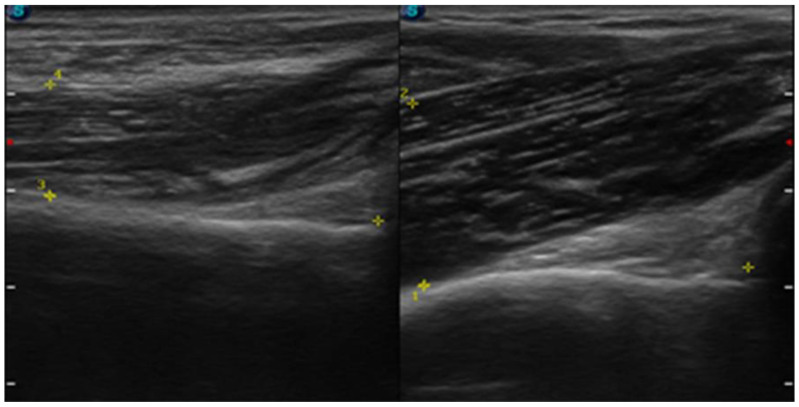
Ultrasound image of a longitudinal section of the brachialis muscle on the affected (**right**) and non-affected (**left**) side (source: personal).

**Table 2 sensors-23-00501-t002:** Summary of electromyography (EMG) data.

Test	Stimulated Points	Latency [ms]	Amplitude [mV]	Duration [ms]	Area [mV × ms]	Stimulation [mA]	Stimulation [ms]	Distance [mm]	Time [ms]	Velocity [m/s]
Right, Musculus biceps brachii, nervus musculocutaneus, C5-C6										
4	Erb’s point	4.3	14.1	6.2	43.3	60	0.1	250	-	-
Right, Musculus brachialis, nervus musculocutaneus, C5-C6										
6	1	5.0	14.5	10.5	37.2	60	0.1	240	-	-
Right, Musculus Triceps brachii, nervus radialis, C6-Th1										
5	1	4.6				60	0.1	250	-	-
Right, Musculus Abductor digiti minimi, nervus ulnaris, C6-Th1										
2	Wrist	2.5	7.0	6.45	20.9	50	0.1	55	-	-
	Elbow	7.2	6.6	6.7	23.5	50	0.1	280	4.7	59.6

[ms]—millisecond; [mV]—millivolt; [mA]—milliampere; [mm]—millimetre; [m/s]—metre per second.

**Table 3 sensors-23-00501-t003:** F-wave parameters—right, abductor digiti minimi, Ulnaris, C8-T1.

	Min.	Max.	Mean	Disparity	Disp. Factor [%]
Lat. [ms]	23.8	26.6	25.0	2.78	11.1
Ampl. F [µV]	121	313	218	192	88.2
Ampl. F/M [%]	0.804	2.09	1.45		
V prox. [m/s]	55.6	65.7	61.7	10.2	16.6

Lat.—latency; Ampl. F—F-wave amplitude; Ampl. F/M—F/M ratio (wave-F amplitude to wave-M amplitude ratio); V prox.—the proximal conduction velocity; [ms]—millisecond; [µV]—microvolt; [%]—percent; [m/s]—meters per second; [min.]—minimal latency; [max.]—maximal latency; [mean]—mean latency; [disparity]—maximal latency minus minimal latency.

**Table 4 sensors-23-00501-t004:** Summary of obtained therapeutic effects.

Examination		Pre-Therapy	Post-Therapy
Pain	permanent	3	0
	after exercise	6	0
2PD	affected	33.33 mm	22.43 mm
	non-affected	20.98 mm	21.44 mm
CSA	affected	0.09 cm^2^	0.06 cm^2^
	non-affected	0.06 cm^2^	0.06 cm^2^
THss BM	affected	11.61 mm	11.77 mm
	non-affected	18.79 mm	18.74 mm
DASH		76 pkt.	43 pkt.
DASH “sport”		16 pkt.	8 pkt.
SF-36	PF	75%	85%
RF		25%	75%
BP		45%	67.5%
GH		45.8%	58.3%
VT		50%	70%
SF		87.5%	100%
RE		66.6%	100%
MH		68%	80%
PCS		47.7%	71.4%
MCS		68%	87.5%

2PD—two-point discrimination; CSA—cross-section area; THss BM—thickness brachialis muscle; DASH—disabilities of the arm, shoulder, and hand; SF-36—36-item short form health survey; PF—physical functioning; RF—role limitations because of physical health problems; BP—bodily pain; GH—general health perceptions; VT—vitality; SF—social role functioning; RE—role limitations because of emotional problems; ME—mental health; PCS—physical component summary; MCS—mental component summary.

## Data Availability

Data are available on requested.

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
