# Peer review of "Ultrasound Diagnostic and Physiotherapy Approach for a Patient with Parsonage–Turner Syndrome—A Case Report"

_sensors, 2023, doi:10.3390/s23010501_

Round 1

Reviewer 1 Report

Dear Authors.

Congratulations for the case-report which could be useful for other clinicians.

There are some comment could be enhancing the presentation and some clarifications claims your attention.

Read the attached document.

Author Response

Congratulations for the case-report which could be useful for other clinicians.

There are some comment could be enhancing the presentation and some clarifications claims your attention.

Response:

Dear reviewer we appreciate the time and effort that you have dedicated to review our manuscript. Thank you for appreciating our work and its clinical importance for clinicians using ultrasound diagnostics and physiotherapy approach based on neurodynamic techniques in the treatment of a rare disease such as PTS. We would like to thank you for your suggestions. We have tried to improve them all to make our manuscript better for our readers.

Methodology

Considering a case report has been presented the Case Report Guidelines (CARE) would be useful for structuring the text.

Response:

In the method part, we added a sentence: „We also checked the entire manuscript using the CARE checklist for case studies to ensure that it contained all the guidelines necessary for this type of work (Table S1).

Table 1. Present the data in a table handmade (better), it looks an screenshot with low quality.

Response:

Thank you for your right attention. We improved the quality of Table 1.

Physiotherapy management At line 121-122 is written no ethics approval was applied, but experimental procedure is described and declared with no evidence with patient´s authorization. Finnally lines 313- 314 is detailed ethics committee. Please clarify.

Response:

Of course, the statement in lines 120-121 is true. There was no consent of the Bioethics Committee because the study was retrospective. This is our editorial error. Line 313-314 has been deleted.

Figure 3. Just affected side is reported, the contralateral would be interesting to understand de image.

Response:

Figure 3 is an MRI scan. Unfortunately, the patient did not have a non-affected side MRI.

Figure 4. The footnote talks about the brachialis muscle, but the trace measurement is about a nerve. Resolve this conflict. By other side these images do not look to have the same protocol. The fascias are not aligned (see the imagen copy-pasted) painted in blue in relation to the bone, painted in pink. It could be reporting different colors and interpretation could be biased. Detail the affected side in the image (A,affected) (B,health) i.e.

Response:

Thank you for your valuable attention. The footnote has been changed. The level of examination of the musculocutaneous nerve was the same. The different picture we see seems to be the result of significant atrophy of the brachialis, which affected the location of the fascia and other structures.

The neurodinamics for all the principal nerves of upperlimb is described, but just ultrasound imaging of musculocutaneus nerve is reported. It do not have sense apply treatment all those neves and just have the evaluation of one. Its not clear, explain.

Response:

Of course, we agree that the use of neurodynamic techniques for all major nerves of the upper limb may have been redundant. PTS, on the other hand, is about the brachial plexus. Since neurodynamic techniques in PTS have not been described before, we decided that the techniques applied to the main nerves would stimulate the brachial plexus to a greater extent.

Discussion

The UI variables are not well contextualized. Strong evidence must be commented around the nerve’s cross-sectional area and muscle thickness in neurological diseases.

Response:

Thank you for your valuable attention. We added a sentence in the discussion.

There is no weight given to the muscle thickness considering the difference reported. This variable did not changed, why do you think there is no change? comment In the discussion section.

Response:

Thank you for your valuable attention. We added a sentence in the discussion.

Conclusion

The muscle thickness is not commented.

Response:

Thank you for your valuable attention. We added a sentence in the conclusion.

Reviewer 2 Report

Ultrasound diagnostic and physiotherapy approach for a patient with Parsonage-Turner syndrome – a case report - The work focuses on a rare and interesting clinical case that can contribute to the diagnosis and treatment of these cases. The work is well structured, the methodology described in a very detailed way with application of scientific language. The presentation of the case is clear, discussion and conclusion seem to me to be in accordance with the proposed objectives. However, I sensitize the authors to improve the quality of the images. Given the rarity, there is not a relevant number of publications on this subject, so considering it adds to the subject area and that the clinical case presents conditions for publication.

Author Response

Ultrasound diagnostic and physiotherapy approach for a patient with Parsonage-Turner syndrome – a case report - The work focuses on a rare and interesting clinical case that can contribute to the diagnosis and treatment of these cases. The work is well structured, the methodology described in a very detailed way with application of scientific language. The presentation of the case is clear, discussion and conclusion seem to me to be in accordance with the proposed objectives. However, I sensitize the authors to improve the quality of the images. Given the rarity, there is not a relevant number of publications on this subject, so considering it adds to the subject area and that the clinical case presents conditions for publication.

Response:

Dear reviewer we appreciate the time and effort that you have dedicated to review our manuscript. Thank you for appreciating our work and its clinical importance for clinicians using ultrasound diagnostics and physiotherapy approach based on neurodynamic techniques in the treatment of a rare disease such as PTS. We would like to thank you for your suggestions. We tried to improve the quality of photos and Table 1.

Reviewer 3 Report

Dear authors, 

Thank you for the opportunity to review your interesting manuscript. The manuscript is well-written and proposes a novel topic in physiotherapy. 

Only there are minor reviews that have to be addressed before considering for publication. Please, considering that you present a case report it will be interesting that the CARE statement will be mentioned in the methods and follow along with the manuscript. Also, consider adding this statement as supplementary material. 

In addition, to help the reader please add more information about outcomes and tools (e.g., SF-36 measured quality of life, and it was scored from 0 to 10 with higher scores indicating higher quality of life). 

Author Response

Thank you for the opportunity to review your interesting manuscript. The manuscript is well-written and proposes a novel topic in physiotherapy. 

Response:

Dear reviewer we appreciate the time and effort that you have dedicated to review our manuscript. Thank you for appreciating our work and its clinical importance for clinicians using ultrasound diagnostics and physiotherapy approach based on neurodynamic techniques in the treatment of a rare disease such as PTS. We would like to thank you for your suggestions. The manuscript has been revised as recommended to make it better for the readers.

Only there are minor reviews that have to be addressed before considering for publication. Please, considering that you present a case report it will be interesting that the CARE statement will be mentioned in the methods and follow along with the manuscript. Also, consider adding this statement as supplementary material. 

Response:

In the method part, we added a sentence: „We also checked the entire manuscript using the CARE checklist for case studies to ensure that it contained all the guidelines necessary for this type of work (Table S1).

In addition, to help the reader please add more information about outcomes and tools (e.g., SF-36 measured quality of life, and it was scored from 0 to 10 with higher scores indicating higher quality of life). 

Response:

Thank you for your suggestion. We've added some important information.

Round 2

Reviewer 1 Report

The document has been improved and becomes in a good alliance for clinicians who can be looking for case-reports.

Good job.